# PEAN: A Diffusion-Based Prior-Enhanced Attention Network for Scene Text Image Super-Resolution

Zuoyan Zhao
School of Computer Science and Engineering,
Southeast University
Key Laboratory of New Generation Artificial Intelligence
Technology and Its Interdisciplinary Applications
(Southeast University), Ministry of Education, China
Nanjing, China
zuoyanzhao@seu.edu.cn

Hui Xue*
School of Computer Science and Engineering,
Southeast University
Key Laboratory of New Generation Artificial Intelligence
Technology and Its Interdisciplinary Applications
(Southeast University), Ministry of Education, China
Nanjing, China
hxue@seu.edu.cn

Pengfei Fang
School of Computer Science and Engineering,
Southeast University
Key Laboratory of New Generation Artificial Intelligence
Technology and Its Interdisciplinary Applications
(Southeast University), Ministry of Education, China
Nanjing, China
fangpengfei@seu.edu.cn

Shipeng Zhu
School of Computer Science and Engineering,
Southeast University
Key Laboratory of New Generation Artificial Intelligence
Technology and Its Interdisciplinary Applications
(Southeast University), Ministry of Education, China
Nanjing, China
shipengzhu@seu.edu.cn

## ABSTRACT

Scene text image super-resolution (STISR) aims at simultaneously increasing the resolution and readability of low-resolution scene text images, thus boosting the performance of the downstream recognition task. Two factors in scene text images, visual structure and semantic information, affect the recognition performance significantly. To mitigate the effects from these factors, this paper proposes a **P**rior-**E**nhanced **A**ttention **N**etwork (**PEAN**). Specifically, an attention-based modulation module is leveraged to understand scene text images by neatly perceiving the local and global dependence of images, despite the shape of the text. Meanwhile, a diffusion-based module is developed to enhance the text prior, hence offering better guidance for the SR network to generate SR images with higher semantic accuracy. Additionally, a multi-task learning paradigm is employed to optimize the network, enabling the model to generate legible SR images. As a result, PEAN establishes new SOTA results on the TextZoom benchmark. Experiments are also conducted to analyze the importance of the enhanced text prior as a means of improving the performance of the SR network. Code is available at https://github.com/jdfxzzy/PEAN.

## CCS CONCEPTS

• **Computing methodologies** → **Reconstruction**.

* Corresponding author: Hui Xue.

*MM '24, October 28–November 1, 2024, Melbourne, VIC, Australia.*
© 2024 Copyright held by the owner/author(s). Publication rights licensed to ACM.
ACM ISBN 979-8-4007-0686-8/24/10
https://doi.org/10.1145/3664647.3680974

## KEYWORDS

Scene Text Image, Super-Resolution, Vision Backbone, Diffusion Models

**ACM Reference Format:**
Zuoyan Zhao, Hui Xue, Pengfei Fang, and Shipeng Zhu. 2024. PEAN: A Diffusion-Based Prior-Enhanced Attention Network for Scene Text Image Super-Resolution. In *Proceedings of the 32nd ACM International Conference on Multimedia (MM '24), October 28–November 1, 2024, Melbourne, VIC, Australia.* ACM, New York, NY, USA, 18 pages. https://doi.org/10.1145/3664647.3680974

## 1 INTRODUCTION

Scene text recognition (STR) focuses on extracting text from images, which has been widely applied in automatic driving [46], intelligent transportation [1], *etc.* However, in real-world applications, a variety of reasons result in captured images being low-resolution (LR), such as the quality of the lens, motion blur, and shaking when capturing photos, leading to blurred text in images. To better read text from such images, researchers formulate the STISR task to reconstruct missing text details in LR images, as a pre-processing step for STR.

For scene text images, two crucial factors determine whether they could be correctly recognized, *i.e.*, visual structure and semantic information [14, 69]. Early attempts at STISR concentrate on adequately recovering the visual structure of LR scene text images [13, 62, 74]. Composed of several CNN-BiLSTM layers, these methods can learn from paired LR-HR images to improve the resolution and readability of scene text images simultaneously. However, the performance is limited due to the fact that they ignore the semantic information of scene text images. This factor has been utilized in recent advancements. These works observe that semantic information plays an important role in guiding the restoration of correct visual structure, and propose numerous text

**Figure 1: Comparison between previous text prior-based STISR methods (row (b, c)) and PEAN. The incorporation of AMM enables PEAN to restore the visual structure of lengthy text in images. However, its performance is limited by the absence of semantic information (row (d)). The introduction of TP-LR partially addresses this limitation, yet its efficacy remains inadequate, leading to several failure cases (row (e)). Considering that TP-HR is a robust alternative, we conduct an exploratory experiment by substituting TP-HR with TP-LR, resulting in superior performance (row (f)). This inspires us to design a module for enhancing the TP-LR so as to obtain the ETP, which demonstrates comparable effectiveness to TP-HR in guiding the SR process (row (g)).**

prior-based methods [19, 34, 35, 75]. That is, the text prior, generated by pre-trained STR models, is leveraged to facilitate the SR process [19, 34, 35, 75], thereby generating correct characters of text in SR images.

Despite improved performance achieved by these approaches, the dominance of visual structure and semantic information persists, as two critical issues in previous studies remain unresolved. Firstly, previous STISR methods [5, 6, 35, 62, 74, 75] rely on Sequential Residual Blocks (SRB) to extract visual features. This module, containing several CNN-BiLSTM layers, *has difficulty in restoring the complete visual structure of images containing long or deformed text string due to its inherent demerits*, *i.e.*, the performance bottleneck of capturing long-range dependence [12, 44]. Secondly, *the introduction of the primary text prior, originating from the interference of low-quality images on recognizers, prevents the SR network from generating images that contain correct semantic information*. Recently, C3-STISR [75] has employed a language model [14] into STISR, utilizing its learned linguistic knowledge to rectify the text prior. Although the rectified prior demonstrates some effectiveness, it lacks sufficient strength in guiding the SR network to produce images with high semantic accuracy.

We propose a **P**rior-**E**nhanced **A**ttention **N**etwork (**PEAN**) to tackle issues caused by the two factors. To begin with, an Attention-based Modulation Module (AMM) is proposed to substitute the SRB, endowing the network with a larger receptive field to images, thereby restoring the visual structure of images with text in various shapes and lengths (shown in Figure 1(d)). Horizontal

and vertical strip-wise attention mechanisms [11, 21, 58] are employed in AMM. Among them, the horizontal attention mechanism can capture the dependence between characters while the vertical attention mechanism can capture the structural information within a character [74]. However, the lack of semantic information limits the capability of such model. As demonstrated by previous works [8, 10], leveraging strong prior information to restrict the solution space plays a vital role in SR problems. Notably, the text prior derived from high-resolution (HR) images is a robust choice for STISR, in view of the high recognition accuracy of HR images. Consequently, we conduct an exploratory experiment wherein we substitute the text prior from LR images (TP-LR) with the text prior from HR images (TP-HR) within such model, yielding superior outcomes (see Figures 1(e) and (f) for comparison, details can be found in § 4.4.1). This inspires the design of a module for enhancing the primary text prior, resulting in the creation of the Enhanced Text Prior (ETP), which is comparable in effectiveness to TP-HR (shown in Figures 1(f) and (g)). The ETP provides valuable guidance to the SR network, promoting the generation of SR images with high semantic accuracy. Given the remarkable performance of diffusion models [20, 56], we propose a diffusion-based Text Prior Enhancement Module (TPEM) to obtain the ETP owing to their ability to map complex distributions [67]. In addition, considering that the goal of STISR is to increase the resolution and readability of LR scene text images, we adopt the Multi-Task Learning (MTL) paradigm in the training phase, where the image restoration task aims at generating high-quality SR images, and the text recognition task stimulates the model to generate more readable SR results. In a nutshell, main contributions of our work are three-fold:

- We devise an AMM containing horizontal and vertical attention mechanisms to model the long-range dependence in scene text images, thereby recovering the visual structure of images with long or deformed text.
- A diffusion-based TPEM is further proposed to enhance the primary text prior. The resulting ETP guides the SR network to generate SR images with improved semantic accuracy.
- Empirical studies show that PEAN attains the SOTA performance on the TextZoom [62] benchmark. We also conduct experiments to explore the reasons behind the performance improvement of the SR network.

## 2 RELATED WORK

### 2.1 Scene Text Image Super-Resolution

Scene text image super-resolution (STISR) has received surging attention in the computer vision community. Different from the classic single image super-resolution (SISR) task, STISR aims at increasing the resolution and legibility of scene text images simultaneously [81], serving as a pre-processing method for the downstream recognition task.

The milestone works in STISR are the TextZoom benchmark and the TSRN model [62], which promote the development of follow-up approaches. We roughly classify them into two categories. One category of methods focuses on recovering the visual structure of LR scene text images. Among them, TSRN [62] and PCAN [74] use several CNN-BiLSTM blocks to complete the SR process. TSAN [82] adopts a gradient-based graph attention method to extract more

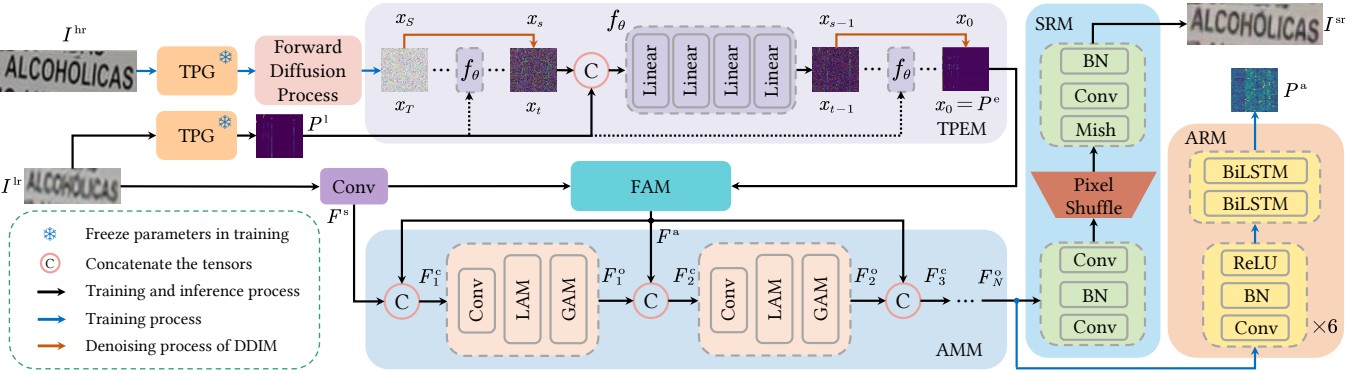

**Figure 2: Overview of the architecture of our proposed Prior-Enhanced Attention Network (PEAN).**

effective representations for STISR. Another category considers the semantic information as the text prior to guide the SR process. In this category, TPGSR [34] and TATT [35] utilize the pre-trained recognizer to generate the text prior from LR images, boosting the model to generate SR images with correct text. C3-STISR [75] employs a language model [14] to rectify the text prior and uses triple clues to realize STISR.

## 2.2 Scene Text Recognition

Scene text recognition (STR) aims at reading text contents from natural images. The pioneering work CRNN [52] uses the CNN-BiLSTM framework and the CTC [18] loss to perform STR for the first time. ASTER [53] further exploits the Thin-Plate Spline (TPS) transformation [22] to understand scene text images with deformed or irregular layouts. The concurrent work MORAN [33] also handles these cases via a multi-object rectification network.

Recently, language models have been integrated into STR models and the fusion of vision and language features shows a great potential to improve the scene text understanding. SRN [69] employs an autoregressive language model to rectify the recognition results generated by visual features. Additionally, ABINet [14] shows that masked language models [9], capable of providing bidirectional representations, constitute another effective option for rectification. Furthermore, PARSeq [2] creatively adopts an internal language model as a spell-checker, eliminating the need for the pre-training process in ABINet [14].

## 2.3 Diffusion Models

In computer vision, diffusion models [20] emerge as robust probabilistic generative models, facilitating tasks like image synthesis [17, 48], text-to-image synthesis [38, 50], image restoration [15, 67] and image inpainting [80] through the iterative diffusion of information among pixels. Recently, diffusion models have also been employed in the super-resolution task. SR3 [51] is the pioneering work that applies diffusion models to SISR. TextDiff [29] represents the initial attempt at a diffusion-based model designed specifically for STISR, focusing on enhancing the visual structure of text within images by refining their contours for a more natural appearance.

In contrast to existing methods, the proposed PEAN uses the diffusion-based TPEM to provide the SR network with enhanced

semantic guidance, further resulting in SR images with heightened semantic accuracy.

## 3 METHODOLOGY

This section first gives an overview of the Prior-Enhanced Attention Network (PEAN). Then we present the proposed Text Prior Enhancement Module (TPEM), Attention-based Modulation Module (AMM) and the Multi-Task Learning (MTL) paradigm.

### 3.1 Overall Architecture

The pipeline of our proposed PEAN is shown in Figure 2. Given one LR image $I^{lr} \in \mathbb{R}^{H \times W \times C}$, the Text Prior Generator (TPG) outputs the recognition probability sequence as the primary text prior $P^l$. Then, the diffusion-based TPEM refines it to obtain the ETP denoted as $P^e$, which can assist the SR network to generate SR images with improved semantic accuracy. Concurrently, a convolutional layer is adopted to extract the shallow visual feature $F^s$ from $I^{lr}$, which is then aligned with the refined text prior by a Feature Alignment Module (FAM). Then an AMM with $N$ blocks is introduced to mine the internal dependence between characters in the image, thereby facilitating the SR process. For the $i^{th}$ block of AMM (i.e., $B_i$), its output $F^o_i$ is firstly concatenated with the aligned feature (i.e., $F^a$) in the channel dimension to get $F^c_{i+1}$. The fusion feature is then sent into $B_{i+1}$ for further processing. Finally, a Super-Resolution Module (SRM) containing several convolutional and batch normalization layers, receives $F^o_N$ as input and utilizes a PixelShuffle [54] operation to generate the SR image $I^{sr} \in \mathbb{R}^{2H \times 2W \times C}$. Notably, in the training phase, $F^o_N$ is also sent into an Auxiliary Recognition Module (ARM), which outputs the recognition probability sequence of the SR image. The outputs of TPEM, SRM and ARM enable the optimization of the model in an Multi-Task Learning (MTL) paradigm, steering the model to generate plausible and readable SR images.

### 3.2 Text Prior Enhancement Module

As demonstrated by previous works [8, 10], strong prior information plays a pivotal role in solving SR problems, while the primary text prior applied in previous works is not powerful enough because it originates from LR scene text images. Our exploratory experiments, detailed in § 4.4.1, also underline the influential role of TP-HR in

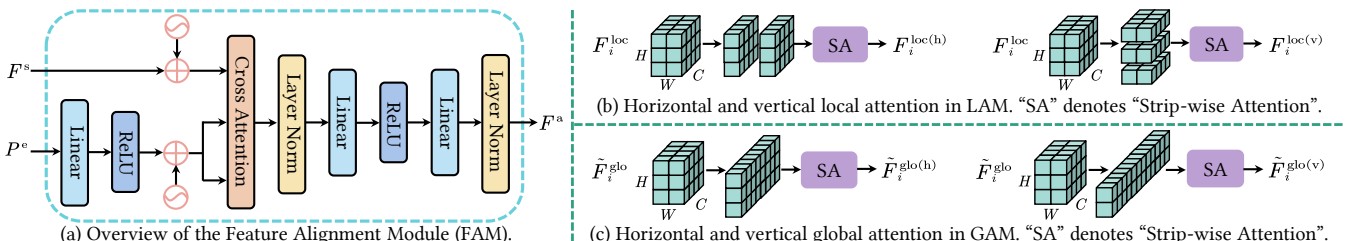

(a) Overview of the Feature Alignment Module (FAM).

(b) Horizontal and vertical local attention in LAM. "SA" denotes "Strip-wise Attention".

(c) Horizontal and vertical global attention in GAM. "SA" denotes "Strip-wise Attention".

**Figure 3: Overview of the architecture of the FAM and the strip-wise attention mechanism inside LAM and GAM.**

guiding the SR network to generate images with improved semantic accuracy for the model. Therefore, we introduce the TPEM to obtain the ETP, which can effectively guide the SR network, similarly to the efficacy of TP-HR. The core component of TPEM is the denoiser, denoted as $f_\theta$, which leverages the reverse diffusion process [20] to estimate the enhanced prior, providing substantial semantic guidance to the SR network.

*3.2.1 Forward Diffusion Process.* In the training phase, with a given HR image, denoted as $I^{hr} \in \mathbb{R}^{2H \times 2W \times C}$, the TPG generates a sequence of recognition probabilities, referred to as $P^h \in \mathbb{R}^{L \times |\mathcal{A}|}$, serving as our ground truth. $L$ is the length of the sequence and $|\mathcal{A}|$ is the cardinality of the recognizable letter set. Consequently, in line with Ho *et al.* [20], we incrementally introduce Gaussian noise denoted as $\epsilon$ to the initial variable $x_0 = P^h$ based on the timestamp, as follows:

$$q\left(x_t \mid x_{t-1}\right) = \mathcal{N}\left(x_t; \sqrt{\alpha_t} x_{t-1}, (1 - \alpha_t)\,\mathbf{I}\right), \quad (1)$$

Here, $\alpha_t$ is a hyperparameter that controls the variance of the added Gaussian noise at each time step. Leveraging the reparameterization trick [24], we can express $x_t$ as:

$$x_t = \sqrt{\bar{\alpha}_t} x_0 + \sqrt{1 - \bar{\alpha}_t} \epsilon, \quad \epsilon \sim \mathcal{N}(\mathbf{0}, \mathbf{I}), \quad (2)$$

where $\alpha_i \in [0, 1]$, $\bar{\alpha}_t = \prod_{i=0}^t \alpha_i$, $t = 1, 2, \cdots, T$. As $T \rightarrow \infty$, $x_T$ converges to an isotropic Gaussian distribution. Consequently, during the inference phase, the forward diffusion process simplifies to initializing $x_T \sim \mathcal{N}(\mathbf{0}, \mathbf{I})$.

*3.2.2 Reverse Diffusion Process.* In the reverse diffusion process, Gaussian noise gradually transforms into the ETP, denoted as $P^e \in \mathbb{R}^{L \times |\mathcal{A}|}$, conditioned on the primary text prior $P^l \in \mathbb{R}^{L \times |\mathcal{A}|}$. The latter is the output recognition probability sequence of the TPG, with $I^{lr}$ as input. This process can be formulated as follows:

$$p_\theta\left(x_{t-1} \mid x_t, P^l\right) = q\left(x_{t-1} \mid x_t, f_\theta\left(x_t, P^l, t\right)\right), \quad (3)$$

where $f_\theta$ is an MLP-based denoising network, the architecture of which can be found in § A of the Supplementary Material. Similar to previous works [29, 67, 68, 76], we opt to directly estimate $P^e$ instead of $\epsilon$ for performance improvement. Experiments in § 4.4.1 verifies the effectiveness of this design. The whole process is supervised by the MAE and CTC loss [18], given by:

$$\mathcal{L}_{\text{diff}} = \lambda_1 \underbrace{\left\| P^h - P^e \right\|_1}_{\mathcal{L}_{\text{mae}}} + \lambda_2 \mathcal{L}_{\text{ctc}}^t, \quad (4)$$

where $\lambda_1$ and $\lambda_2$ are the weight of the two losses. During the training phase, this reverse sampling process is a Markov process containing $T$ steps, which is computationally intensive. Therefore, in the inference phase, we adopt the sampling strategy of DDIM [56] with $S$ steps ($S \ll T$) following previous works [80]. Experiments in the Supplementary Material validate the effectiveness and efficiency of this design.

Besides, a feature alignment process between the shallow visual feature and the ETP is required to facilitate the SR process. We design a Feature Alignment Module (FAM) to obtain the aligned feature $F^a \in \mathbb{R}^{H \times W \times C_1}$, where $C_1$ is the dimension of the shallow visual feature. The architecture of the FAM is shown in Figure 3(a).

## 3.3 Attention-Based Modulation Module

We design an Attention-based Modulation Module (AMM) to capture long-range dependence, thereby effectively restoring the visual structure of images with long or deformed text. As can be seen in Figure 2, AMM contains $N$ blocks, each block (*i.e.*, $B_i$) including a simple convolutional layer, a Local Attention Module (LAM) and a Global Attention Module (GAM). For $B_i$, it receives the output of $B_{i-1}$, which is then concatenated with $F^a$ channel-wisely to get $F_i^c$, serving as the input of this block. Since the dimension of $F_i^c$ is located in $\mathbb{R}^{H \times W \times 2C_1}$, a convolutional layer is adopted to project it to the same space as $F_{i-1}^o$, as:

$$F_i^{\text{loc}} = \text{Conv}\left(\text{Concat}\left(F_{i-1}^o, F^a\right)\right) \in \mathbb{R}^{H \times W \times C_1}. \quad (5)$$

Firstly, a LAM is adopted to model the local similarity between intra- and inter-character features. Considering that for a scene text image, the horizontal contexts contain correlation information between characters while the vertical contexts contain internal features inside a character, such as the stroke information [62, 74], we propose to perform the strip-wise attention mechanism [11, 21, 58] on the fusion feature to capture long-range dependence. Taking the horizontal attention as an example. As shown in Figure 3(b), $F_i^{\text{loc}}$ is split into $W$ strips in the width dimension and the attention mechanism is applied to each of the $W$ strips. The resulting feature is then concatenated to form the horizontal feature $F_i^{\text{loc(h)}} \in \mathbb{R}^{H \times W \times C_1}$. Likewise, the vertical attention mechanism can also result in the vertical feature $F_i^{\text{loc(v)}} \in \mathbb{R}^{H \times W \times C_1}$. $F_i^{\text{loc(h)}}$ and $F_i^{\text{loc(v)}}$ are concatenated in the channel dimension and a convolutional layer is used to fuse them. Then the FFN with residual connection is introduced to perform non-linear transformation, given by:

$$F_i^{\text{glo}} = F_i^{\text{loc}} + \text{Conv}\left(\text{Concat}\left(F_i^{\text{loc(h)}}, F_i^{\text{loc(v)}}\right)\right), \quad (6)$$

$$\tilde{F}_i^{\text{glo}} = F_i^{\text{glo}} + \text{FFN}\left(F_i^{\text{glo}}\right) \in \mathbb{R}^{H \times W \times C_1}. \tag{7}$$

However, the interaction between character-level features is limited in a local manner [79]. To encourage global interaction, we employ the strip-wise GAM as the complementation of the LAM, thus further modeling the global similarity between intra- and inter-character features. Specifically, the width dimension of $\tilde{F}_i^{\text{glo}}$ is mer-ged with the channel dimension, leading to a feature map located in $\mathbb{R}^{H \times C_1 W}$, as can be seen in Figure 3(c). The strip-wise attention mechanism is imposed on it to capture the global inter-character dependence, resulting in the feature $\tilde{F}_i^{\text{glo(h)}} \in \mathbb{R}^{H \times C_1 W}$. Likewise, the height dimension of $\tilde{F}_i^{\text{glo}}$ is merged with the channel dimension, and the attention mechanism is applied to this feature map, leading to the feature $\tilde{F}_i^{\text{glo(v)}} \in \mathbb{R}^{W \times C_1 H}$. With the aid of them, the horizontal and vertical attention mechanisms can work globally to promote global-level interaction of character features. Similar workflow as Eq. (6) and (7) is adopted to get the final feature of $B_i$, namely $F_i^{\text{o}} \in \mathbb{R}^{H \times W \times C_1}$.

## 3.4 Multi-Task Learning

While previous researches [19, 34, 35, 70, 75] adopt the MTL paradigm by incorporating an additional text prior loss [35] to fine-tune the TPG for better text priors, our approach differs in the utilization of the MTL paradigm. We employ this paradigm not only to ensure image quality but also to facilitate text recognition. This distinction arises from the fact that STISR aims to simultaneously enhance the resolution and legibility of scene text images [81].

*3.4.1 **Image Restoration Task.*** To begin with, we employ an image restoration task to generate high-quality SR images. As can be seen in Figure 3, the output of the last block of AMM, *i.e.*, $F_N^{\text{o}}$, is sent into a Super-Resolution Module (SRM) to get the SR image $I^{\text{sr}} \in \mathbb{R}^{2H \times 2W \times C}$. It uses several convolutional layers with batch normalization and Mish [37] activation function for feature refinement and adopts a PixelShuffle [54] operation to increase the resolution. To ensure the pixel-level and structure-level coherence between $I^{\text{sr}}$ and $I^{\text{hr}}$, the Mean-Square-Error (MSE) loss and the Stroke-Focused Module (SFM) loss [6] are applied between the two images, formulated as:

$$\mathcal{L}_{\text{img}} = \lambda_3 \underbrace{\left\| I^{\text{hr}} - I^{\text{sr}} \right\|_2^2}_{\mathcal{L}_{\text{mse}}} + \lambda_4 \underbrace{\left\| A^{\text{hr}} - A^{\text{sr}} \right\|_1}_{\mathcal{L}_{\text{sfm}}}, \tag{8}$$

where $A$ is the attention map fetched from a Transformer-based recognizer [6]. $\lambda_3$ and $\lambda_4$ are the weights of the MSE and SFM loss respectively.

*3.4.2 **Text Recognition Task.*** Considering that high image quality may not be equal to superior recognition results [19, 29], we introduce an Auxiliary Recognition Module (ARM), steering the model to generate SR images with promising legibility. In the inference phase, this module is abandoned, causing no extra computation complexity.

We exploit the classic and effective CNN-BiLSTM architecture applied in the STR task to design the ARM for simplicity [52]. Its output is a recognition probability sequence on the basis of $F_N^{\text{o}}$ and we denote it as $P^{\text{a}} \in \mathbb{R}^{L \times |\mathcal{A}|}$, where $L$ is the length of the

sequence and $|\mathcal{A}|$ is the cardinality of the recognizable letter set. The CTC [18] loss denoted as $\mathcal{L}_{\text{ctc}}^{\text{a}}$ is imposed between $P^{\text{a}}$ and the ground truth for better optimization. In a word, the total loss of the text recognition task is:

$$\mathcal{L}_{\text{txt}} = \lambda_5 \mathcal{L}_{\text{ctc}}^{\text{a}}, \tag{9}$$

where $\lambda_5$ is the weight of the CTC [18] loss for the text recognition task. In the training phase, we optimize the parameters of PEAN. The total loss is:

$$\mathcal{L} = \mathcal{L}_{\text{diff}} + \mathcal{L}_{\text{img}} + \mathcal{L}_{\text{txt}}. \tag{10}$$

## 4 EXPERIMENTS

In this section, we first introduce the datasets and evaluation metrics. Then the implementation details are thoroughly described. Subsequently, comprehensive experiments are conducted to demonstrate that our proposed PEAN is an effective alternative for STISR.

### 4.1 Datasets and Evaluation Metrics

We conduct experiments on the TextZoom [62] benchmark. However, due to some inherent drawbacks of it, these experiments are not enough to reflect that PEAN surely has the ability to restore the complete visual structure. Therefore, we select 651 images with the resolution no greater than $16 \times 64$ as LR images from the IIIT5K [36], SVTP [42] and IC15 [23] datasets for evaluation. Details of the datasets and drawbacks of TextZoom can be found in § B.2 of the Supplementary Material. For TextZoom, following previous works [5, 6, 19, 34, 35, 62, 70, 75], we adopt ASTER [53], MORAN [33] and CRNN [52] for evaluation. In § B.1 of the Supplementary Material, we also adopt three recent Transformer-based recognizers, *i.e.*, MGP-STR [61], ABINet [14] and VisionLAN [64] for evaluation. For the constructed dataset, PSNR and SSIM [65] are selected as metrics to evaluate the image quality.

### 4.2 Implementation Details

Our model is implemented with the Pytorch 1.10 deep learning library [41]. All of the experiments are conducted on 1 NVIDIA TITAN RTX GPU. In the training phase, the model is trained for 200 epochs and optimized by the AdamW [31] optimizer. The learning rate and the size of the mini-batch are set as 0.001 and 32 respectively. The weight of each loss is set as $\lambda_1 = 1$, $\lambda_2 = 1$, $\lambda_3 = 0.8$, $\lambda_4 = 75$, $\lambda_5 = 1$. In terms of the network architecture, the number of blocks in AMM is 6. The sampling timestep in the TPEM is set as 1. Following CRNN [52], the number of convolutional layers applied in the ARM is 6. We first drop the TPEM and pre-train the model with TP-HR, then the TPEM is introduced and weights of parameters obtained by pre-training are initialized to continue the fine-tuning process of the model. Of note, if this setting is abandoned, PEAN can still achieve SOTA performance on the TextZoom [62] dataset. Ablation studies about the weight of loss functions and the pre-training and fine-tuning setting can be found in § B.3.5 and § B.3.6 of the Supplementary Material respectively. In the main paper, we study our method on top of PARSeq [2] as the TPG. In § B.3.6 of the Supplementary Material, we also study the cases when CRNN [52] and ABINet [14] are adopted as the TPG.

**Table 1: The recognition accuracy of some mainstream STISR methods on the three subsets of TextZoom. Best scores are bold.**

| Methods | Accuracy of ASTER [53] (%) | | | | Accuracy of MORAN [33] (%) | | | | Accuracy of CRNN [52] (%) | | | |
|---|---|---|---|---|---|---|---|---|---|---|---|---|
| | Easy | Medium | Hard | Average | Easy | Medium | Hard | Average | Easy | Medium | Hard | Average |
| LR | 62.4 | 42.7 | 31.6 | 46.6 | 59.4 | 36.0 | 28.2 | 42.3 | 37.5 | 21.2 | 21.4 | 27.3 |
| SRCNN [10] | 69.4 | 43.4 | 32.2 | 49.5 | 63.2 | 39.0 | 30.2 | 45.3 | 38.7 | 21.6 | 20.9 | 27.7 |
| SRResNet [27] | 69.6 | 47.6 | 34.3 | 51.3 | 60.7 | 42.9 | 32.6 | 46.3 | 39.7 | 27.6 | 22.7 | 30.6 |
| RDN [73] | 70.0 | 47.0 | 34.0 | 51.5 | 61.7 | 42.0 | 31.6 | 46.1 | 41.6 | 24.4 | 23.5 | 30.5 |
| RRDB [63] | 70.9 | 44.4 | 32.5 | 50.6 | 63.9 | 41.0 | 30.8 | 46.3 | 40.6 | 22.1 | 21.9 | 28.9 |
| LapSRN [26] | 71.5 | 48.6 | 35.2 | 53.0 | 64.6 | 44.9 | 32.2 | 48.3 | 46.1 | 27.9 | 23.6 | 33.3 |
| ESRT [10] | 69.8 | 49.1 | 35.2 | 52.5 | 61.9 | 41.7 | 32.2 | 46.3 | 48.2 | 27.9 | 25.8 | 34.8 |
| Omni-SR [60] | 71.2 | 52.3 | 38.1 | 54.9 | 66.7 | 47.9 | 36.5 | 51.4 | 54.8 | 37.4 | 29.4 | 41.4 |
| SRFormer [78] | 69.0 | 45.1 | 32.8 | 50.2 | 61.3 | 39.6 | 29.9 | 44.7 | 41.0 | 22.8 | 22.9 | 29.6 |
| TSRN [62] | 75.1 | 56.3 | 40.1 | 58.3 | 70.1 | 53.3 | 37.9 | 54.8 | 52.5 | 38.2 | 31.4 | 41.4 |
| TBSRN [5] | 75.7 | 59.9 | 41.6 | 60.1 | 74.1 | 57.0 | 40.8 | 58.4 | 59.6 | 47.1 | 35.3 | 48.1 |
| PCAN [74] | 77.5 | 60.7 | 43.1 | 61.5 | 73.7 | 57.6 | 41.0 | 58.5 | 59.6 | 45.4 | 34.8 | 47.4 |
| TG [6] | 77.9 | 60.2 | 42.4 | 61.3 | 75.8 | 57.8 | 41.4 | 59.4 | 61.2 | 47.6 | 35.5 | 48.9 |
| SGENet [57] | 75.8 | 60.7 | 45.0 | 61.4 | 71.5 | 56.2 | 41.4 | 57.3 | 59.4 | 47.9 | 37.7 | 49.0 |
| TPGSR [34] | 78.9 | 62.7 | 44.5 | 62.8 | 74.9 | 60.5 | 44.1 | 60.5 | 63.1 | 52.0 | 38.6 | 51.8 |
| TATT [35] | 78.9 | 63.4 | 45.4 | 63.6 | 72.5 | 60.2 | 43.1 | 59.5 | 62.6 | 53.4 | 39.8 | 52.6 |
| C3-STISR [75] | 79.1 | 63.3 | 46.8 | 64.1 | 74.2 | 61.0 | 43.2 | 60.5 | 65.2 | 53.6 | 39.8 | 53.7 |
| TATT + DPMN [81] | 79.3 | 64.1 | 45.2 | 63.9 | 73.3 | 61.5 | 43.9 | 60.4 | 64.4 | 54.2 | 39.2 | 53.4 |
| TSAN [82] | 79.6 | 64.1 | 45.3 | 64.1 | 78.4 | 61.3 | 45.1 | 62.7 | 64.6 | 53.3 | 38.8 | 53.0 |
| TEAN [55] | 80.4 | 64.5 | 45.6 | 64.6 | 76.8 | 60.8 | 43.4 | 61.4 | 63.7 | 52.5 | 38.1 | 52.2 |
| MSPIE [83] | 80.4 | 63.4 | 46.3 | 64.4 | 74.0 | 61.4 | 44.4 | 60.8 | 64.5 | 54.2 | 39.6 | 53.5 |
| TCDM [39] | 81.3 | 65.1 | 50.1 | 65.5 | 77.6 | 62.9 | 45.9 | 62.2 | 67.3 | 57.3 | 42.7 | 55.7 |
| LEMMA [19] | 81.1 | 66.3 | 47.4 | 66.0 | 77.7 | 64.4 | 44.6 | 63.2 | 67.1 | 58.8 | 40.6 | 56.3 |
| RTSRN [70] | 80.4 | 66.1 | 49.1 | 66.2 | 77.1 | 63.3 | 46.5 | 63.2 | 67.0 | 59.2 | 42.6 | 57.0 |
| RGDiffSR [77] | 81.1 | 65.4 | 49.1 | 66.2 | 78.6 | 62.1 | 45.4 | 63.1 | 67.6 | 56.5 | 42.7 | 56.4 |
| TextDiff [29] | 80.8 | 66.5 | 48.7 | 66.4 | 77.7 | 62.5 | 44.6 | 62.7 | 64.8 | 55.4 | 39.9 | 54.2 |
| PEAN | **84.5** | **71.4** | **52.9** | **70.6** | **79.4** | **67.0** | **49.1** | **66.1** | **68.9** | **60.2** | **45.9** | **59.0** |
| HR | 94.2 | 87.7 | 76.2 | 86.6 | 91.2 | 85.3 | 74.2 | 84.1 | 76.4 | 75.1 | 64.6 | 72.4 |

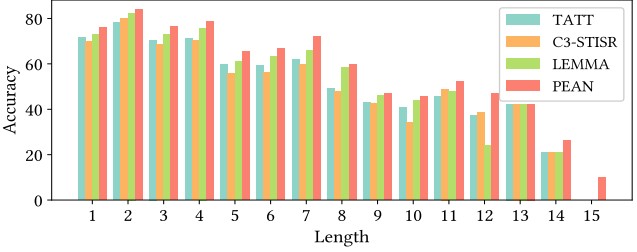

**Figure 4: Statistics on the performance of different text prior-based models with publicly available weights on images containing text of different lengths.**

## 4.3 Comparing with State-of-the-Art Methods

We evaluate our proposed PEAN on the TextZoom benchmark [62] and compare its performance with several typical STISR methods. The results are presented in Table 1. It is evident that our proposed PEAN achieves new SOTA performance with significant improvements. For instance, when considering the recognition accuracy of ASTER [53], our model shows an average improvement of **+4.2**

compared with the existing SOTA model, namely, TextDiff [29]. Furthermore, we provide statistics on the performance of different text prior-based models [19, 35, 75] on images containing text of different lengths. For a fair comparison, all selected models use publicly available weights. As illustrated in Figure 4, PEAN outperforms other models across nearly every text length. It is particularly superior to previous works with images containing lengthy text, establishing itself as the first model capable of processing images containing text with up to 15 letters.

In addition, we provide some visualizations of samples from TextZoom recovered by several representative STISR models, as shown in Figure 5. With the assistance of the ETP, PEAN is able to generate SR images with improved semantic accuracy compared with previous text prior-based methods [19, 35, 75]. Although early methods such as TSRN [62] can generate SR images with correct semantic information for words like "cooking", it is obvious that the visual structure of the text in images is disastrous.

Additionally, we perform experiments on the dataset we built without retraining the models. Improved performance on PSNR and SSIM in Table 2 shows that compared with previous representative STISR methods, PEAN is better at recovering the visual structure of the text in images.

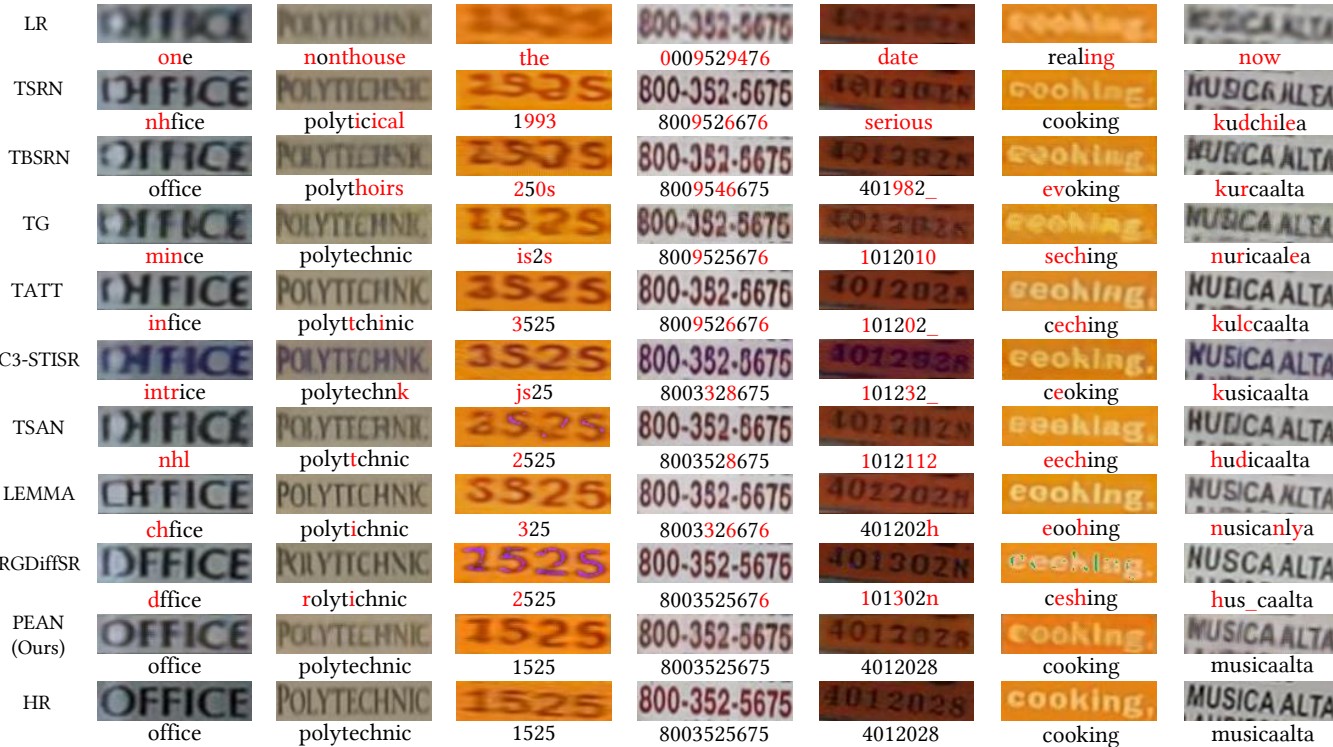

**Figure 5: Visualization of SR images and their recognition results by ASTER. Red characters indicate wrong recognition results.**

**Table 2: The performance of representative STISR models on the dataset we built.**

| Methods | PSNR | SSIM |
| --- | --- | --- |
| TSRN [62] | 21.48 | 0.7743 |
| TBSRN [5] | 23.01 | 0.7967 |
| TG [6] | 21.84 | 0.7116 |
| TATT [35] | 23.31 | 0.8012 |
| C3-STISR [75] | 21.03 | 0.7602 |
| LEMMA [19] | 22.09 | 0.7549 |
| PEAN | **24.24** | **0.8021** |

## 4.4 Ablation Study

Here we conduct ablation studies to demonstrate the effectiveness of components in our model. All the experiments are conducted on TextZoom and we report the recognition accuracy of ASTER [53].

*4.4.1 Text Prior and the Enhancement Module.* We conduct experiments to verify the importance of incorporating the text prior and the enhancement module into our model. The results shown in Table 3 justify that the introduction of the text prior can improve the performance of the SR process. However, this primary text prior from LR images is not robust enough to guide the SR network to generate SR images with high semantic accuracy. Considering that TP-HR can serve as a powerful alternative for guidance, we conduct an exploratory experiment wherein we substitute TP-LR

with TP-HR. Results shown in the last row of the table demonstrate that leveraging the powerful prior information from HR images can significantly boost the performance of text prior-based STISR methods. This observation motivates us to design a module aimed at enhancing the primary text prior from LR images, thereby providing further guidance to the SR network for generating images with high semantic accuracy. Results displayed in the table indicate that our proposed TPEM is able to produce the ETP, which significantly improves the performance over the model with TP-LR among all subsets by **+6.8** on average.

**Table 3: Analysis of the impact of the text prior and the TPEM. The last row shows the results of the exploratory experiment as illustrated in Figure 1(f).**

| TP-LR | TPEM | TP-HR | Easy | Medium | Hard | Average |
| --- | --- | --- | --- | --- | --- | --- |
| | | | 75.7 | 60.2 | 42.1 | 60.4 |
| ✓ | | | 79.7 | 62.3 | 46.1 | 63.8 |
| ✓ | ✓ | | **84.5** | **71.4** | **52.9** | **70.6** |
| | | ✓ | 88.4 | 75.5 | 61.3 | 75.9 |

*4.4.2 Impact of the AMM.* Here we perform experiments to justify the superiority of the AMM adopted in our model. As presented in Table 4, the comparison reveals the following points: (1) Our proposed AMM handles the STISR task in a more effective way than the CNN-BiLSTM-based SRB [62]. The local inductive bias of the CNN [12, 44] and the rigid nature of BiLSTM [59, 81]

limit its performance in recovering the visual structure of images with long or deformed text [5, 35], while the AMM can solve this problem. (2) The typical ViT-based architectures [11, 12, 30] show trivial performance, as a result of the tendentious design for high-level vision tasks, which neglects the inherent characteristic of STISR. (3) Although employing the strip-wise attention mechanism, Stripformer [58], which also leverages the conditional positional encodings [7] and several residual blocks for the image deblurring task, fails in the STISR task. On average, PEAN obtains an improvement of the performance of **+5.9** from the AMM. Experiments in § B.3.4 of the Supplementary Material show that even without the MTL paradigm, the AMM still outperforms the SRB [62].

**Table 4: Ablation study on the impact of the AMM.**

| Methods | Easy | Medium | Hard | Average |
|---|---|---|---|---|
| SRB [62] | 80.1 | 64.4 | 46.4 | 64.7 |
| ViT [12] | 81.8 | 65.7 | 49.5 | 66.7 |
| Swin [30] | 73.8 | 55.1 | 39.0 | 57.1 |
| CSWin [11] | 70.2 | 52.9 | 37.2 | 54.5 |
| Stripformer [58] | 72.9 | 53.6 | 37.3 | 55.7 |
| AMM | **84.5** | **71.4** | **52.9** | **70.6** |

*4.4.3* ***Effect of the MTL Paradigm***. In this part, we conduct experiments to show the effect of the MTL paradigm introduced in our work. The results shown in Table 5 indicate that: (1) The stroke-based SFM loss [6] provides an average contribution of **+4.6** for the improved performance, which convinces us that the preservation of visual structure plays a vital role in STISR. (2) The text recognition task employed in our work is also indispensable. The CTC [18] loss applied on the output of ARM ($\mathcal{L}_{ctc}^{a}$) brings correct semantic constraint for the AMM. Besides, the CTC loss imposed on the ETP ($\mathcal{L}_{ctc}^{t}$) ensures the coherence between the text prior and the ground truth text label, guiding the training of the enhancement module in a precise way. We provide more detailed experiments in § B.3.4 of the Supplementary Material.

**Table 5: Ablation study on the effect of MTL.**

| Loss Functions | Easy | Medium | Hard | Average |
|---|---|---|---|---|
| $\mathcal{L}_{mse}$ | 76.2 | 58.8 | 41.5 | 59.9 |
| $+\ \mathcal{L}_{sfm}$ | 79.2 | 64.3 | 47.0 | 64.5 |
| $+\ \mathcal{L}_{mae}$ | 79.6 | 65.1 | 47.1 | 64.9 |
| $+\ \mathcal{L}_{ctc}^{t}$ | 81.4 | 68.8 | 50.7 | 67.9 |
| $+\ \mathcal{L}_{ctc}^{a}$ | **84.5** | **71.4** | **52.9** | **70.6** |

## 4.5 Representation Analysis

In this section, we delve into the reasons about the ability of the ETP that guides the SR network in generating images with improved semantic accuracy. In Table 3, we observe superior performance for PEAN with TP-HR, while PEAN with TP-LR exhibits poorer results compared to PEAN with ETP, as demonstrated in Table 3. To further investigate these observations, following the common

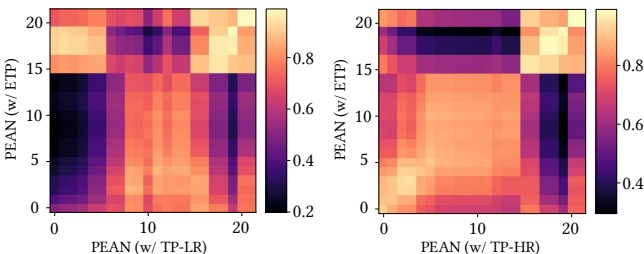

**Figure 6: Results of representation comparison by CKA [25].**

analysis setting on Vision Transformers [40, 44], we employ the linear centered kernel alignment (CKA) [25] to assess the similarity of representations among these three models.

As depicted in Figure 2, the introduction of the text prior influences the representations of the AMM and SRM during the inference phase. Therefore, we focus on comparing the representational similarity of the layers in these two modules. We categorize the 22 layers into two groups for analysis: layers of AMM ($0^{th}$~$11^{th}$ layer) and layers of SRM ($12^{th}$~$21^{st}$ layer). As shown in the diagonal section of Figure 6, distinct differences in representations are evident across all layers of AMM when comparing PEAN (w/ ETP) and PEAN (w/ TP-LR). Conversely, a consistently high similarity is observed between PEAN (w/ ETP) and PEAN (w/ TP-HR). Regarding the layers of SRM, variations in representational similarity mainly exists in the shallow layers ($12^{th}$~$14^{th}$ layer).

Consequently, we can conclude that the power of the ETP lies in its ability to *make the representations learned by AMM and SRM more similar to those learned by the corresponding modules in PEAN (w/ TP-HR)*, which is known for its superior performance. This study further justifies that the combination of our proposed TPEM and AMM brings out powerful capabilities.

## 5 CONCLUSION

In this paper, we propose a Prior-Enhanced Attention Network (PEAN) for scene text image super-resolution (STISR). Specifically, we design a Text Prior Enhancement Module (TPEM) to provide the ETP for the subsequent SR process, enabling SR images to contain accurate semantic information. Moreover, an Attention-based Modulation Module (AMM) is devised to obtain local and global coherence in scene text images, which can recover the visual structure of images with text in various sizes and deformations. Additionally, we introduce the Multi-Task Learning (MTL) paradigm to improve the legibility of LR images. Experiments demonstrate that our proposed PEAN achieves SOTA performance through the interaction of these designs. We believe our work will serve as a strong baseline for future works, and will push forward the research of STISR as well as other sub-fields of scene text images.

## ACKNOWLEDGMENTS

This work was supported by the National Natural Science Foundation of China (Nos. 62076062 and 62306070) and the Social Development Science and Technology Project of Jiangsu Province (No. BE2022811). Furthermore, the work was also supported by the Big Data Computing Center of Southeast University.

# SUPPLEMENTARY MATERIAL

In this Supplementary Material, we provide:

- More details about the MLP-based denoising network, denoted as $f_\theta$, in the TPEM. This is mentioned in the "Methodology" part (§ 3.2.2) of the main paper.
- More experiments and ablation studies on the TextZoom Dataset. This is mentioned in the "Experiments" part (§ 4.4) of the main paper.
- More visualization results on the dataset we built. This is also mentioned in the "Experiments" part (§ 4.1, 4.2) of the main paper.

## A  DETAILS OF THE DENOISING NETWORK

In this section, we present a detailed description of the denoising network, denoted as $f_\theta$, employed in the TPEM. In prevalent diffusion models applied to tasks such as Single Image Super-Resolution (SISR) [16, 51] and image deblurring [47, 66], where the network is designed to process images, researchers often opt for the U-Net architecture [49] as the denoising network. However, in our work, the TPEM is designed to enhance the primary text prior, which is a recognition probability sequence. To achieve this objective, we introduce an MLP-based architecture. The input to $f_\theta$ consists of three components: the noisy text prior at timestep $t$ (denoted as $x_t$), the primary text prior extracted from low-resolution (LR) images (denoted as $P^l$), and the timestep (denoted as $t$). Of these inputs, $P^l$ is concatenated with $x_t$ along the second dimension, serving as a conditioning factor for the denoising process. To reduce the dimension and obtain the fused feature $x_t^0$, we employ a 1D convolutional layer with a kernel size of $1 \times 1$. Simultaneously, the timestep $t$ is encoded into a time embedding (denoted as $t_e$) using a positional encoding module [59]. Subsequently, we utilize four MLP blocks to refine the feature based on the time embedding, with the output of the final MLP layer representing the denoised feature at timestep $t$, which is also the input for timestep $t - 1$. For a comprehensive overview of the architecture of $f_\theta$, please refer to Table I.

## B  MORE EXPERIMENTS ON TEXTZOOM

In this section, we conduct more experiments and ablation studies on the TextZoom benchmark [62] to further demonstrate that our proposed Prior-Enhanced Attention Network (PEAN) can serve as an effective alternative for scene text image super-resolution (STISR). TextZoom [62] is a common benchmark for STISR, containing 17367 and 4373 paired LR-HR images collected in natural scenarios for training and testing, respectively. According to the degree of blurriness, the testing set is divided into three subsets, namely easy (1619 pairs), medium (1411 pairs) and hard (1343 pairs). The sizes of LR and HR images are $16 \times 64$ and $32 \times 128$ respectively. Noteworthy, following the common practice in existing works, in this paper, the reported "Average" results are the weighted average on the three subsets of TextZoom, which is formulated as:

$$Acc_{avg} = \frac{Acc_e \cdot N_e + Acc_m \cdot N_m + Acc_h \cdot N_h}{N_e + N_m + N_h}, \qquad (11)$$

where $Acc_e$, $Acc_m$ and $Acc_h$ denote the recognition accuracy on the "easy", "medium" and "hard" subsets respectively. $N_e$, $N_m$ and

**Table I: Architecture of the MLP-based denoising network.** $N$ is the size of the mini-batch. Grey rows show the components of MLP Block 1, similar to MLP Block 2, 3 and 4.

| Input | Input size | Output | Output size | Module / Operation |
|---|---|---|---|---|
| $x_t$ | $[N, 26, 37]$ | $x_t^0$ | $[N, 52, 37]$ | Concatenate |
| $P^l$ | $[N, 26, 37]$ | | | |
| $x_t^0$ | $[N, 52, 37]$ | $x_t^0$ | $[N, 26, 37]$ | Convolution |
| $t$ | $[N, 1]$ | $t_e$ | $[N, 1, 26]$ | Positional Encoding |
| $x_t^0$ | $[N, 26, 37]$ | $x_t^0$ | $[N, 26, 37]$ | Batch Normalization |
| $x_t^0$ | $[N, 26, 37]$ | $x_t^0$ | $[N, 26, 148]$ | Linear |
| $x_t^0$ | $[N, 26, 148]$ | $x_t^0$ | $[N, 26, 148]$ | Swish [45] Function |
| $t_e$ | $[N, 1, 26]$ | $t_e$ | $[N, 26, 1]$ | Linear & Reshape |
| $x_t^0$ | $[N, 26, 148]$ | $x_t^1$ | $[N, 26, 148]$ | Repeat & Add |
| $t_e$ | $[N, 26, 1]$ | | | |
| $x_t^1$ | $[N, 26, 148]$ | $x_t^2$ | $[N, 26, 296]$ | MLP Block 2 |
| $t_e$ | $[N, 1, 26]$ | | | |
| $x_t^2$ | $[N, 26, 296]$ | $x_t^3$ | $[N, 26, 148]$ | MLP Block 3 |
| $t_e$ | $[N, 1, 26]$ | | | |
| $x_t^3$ | $[N, 26, 148]$ | $x_{t-1}$ | $[N, 26, 37]$ | MLP Block 4 |
| $t_e$ | $[N, 1, 26]$ | | | |

$N_h$ denote the number of images in the corresponding subset. For TextZoom, $N_e = 1619$, $N_m = 1411$, $N_h = 1343$.

### B.1  Recognition Accuracy of SOTA Recognizers

Following the common practice of existing works, in § 4.3 of the main paper, we introduce the recognition accuracy of SR images on three classic scene text recognizers, *i.e.*, ASTER [53], MORAN [33] and CRNN [52] for evaluation, and our proposed PEAN achieves new SOTA results with substantial performance improvement. Here we employ three recent Transformer-based recognizers, *i.e.*, MGP-STR [61], ABINet [14] and VisionLAN [64] for further evaluation to demonstrate the robustness of PEAN. The results are shown in Table II, from which we can conclude that PEAN can still achieve the SOTA performance under the evaluation of recent recognizers. This further justifies that PEAN can surely increase both the resolution and readability of scene text images, regardless of which recognizer we choose for evaluation.

### B.2  Quality of SR Images

Following the common practice of previous works, we use the Peak Signal-to-Noise Ratio (PSNR) and the Structural Similarity Index Measure (SSIM) [65] metrics, which are widely utilized in the classic SISR task, to evaluate the quality of SR images. The results presented in Table III shows that the quality of SR images generated by our proposed PEAN is comparable to existing works.

Different from SISR, which aims at improving the image quality of LR natural images, STISR concentrates more on increasing the readability of scene text images [62, 81]. Therefore, PSNR and SSIM are *NOT* the most suitable metrics for STISR because we empirically find that readability is not closely related to image quality. Firstly, as shown in Figure I, it is common that LR images have higher PSNR or SSIM than SR images. However, it is very difficult for us to distinguish text in images if we are given such LR images, but SR images generated by our proposed PEAN make this task

**Table II: The recognition accuracy of some mainstream STISR methods by three recent scene text recognizers on the three subsets of TextZoom. The best scores are shown in bold. Note that as to methods used for comparison, we adopt the pre-trained model released by their authors for evaluation.**

| Methods | Accuracy of MGP-STR [61] (%) | | | | Accuracy of ABINet [14] (%) | | | | Accuracy of VisionLAN [64] (%) | | | |
|---|---|---|---|---|---|---|---|---|---|---|---|---|
| | Easy | Medium | Hard | Average | Easy | Medium | Hard | Average | Easy | Medium | Hard | Average |
| LR | 73.4 | 59.9 | 45.9 | 60.6 | 77.4 | 58.4 | 43.5 | 60.9 | 74.6 | 53.3 | 39.5 | 56.9 |
| TSRN [62] | 67.3 | 58.7 | 42.7 | 57.0 | 76.2 | 61.4 | 44.8 | 61.8 | 75.2 | 58.3 | 42.7 | 59.8 |
| TBSRN [5] | 72.3 | 62.9 | 45.9 | 61.2 | 80.2 | 65.6 | 48.3 | 65.7 | 78.1 | 62.7 | 45.3 | 63.1 |
| TG [6] | 71.7 | 64.7 | 46.3 | 61.6 | 79.8 | 67.1 | 49.1 | 66.3 | 78.2 | 63.9 | 44.3 | 63.2 |
| TATT [35] | 71.3 | 61.7 | 45.9 | 60.4 | 81.0 | 65.8 | 50.0 | 66.6 | 79.7 | 63.9 | 47.8 | 64.8 |
| C3-STISR [75] | 73.6 | 63.3 | 47.8 | 62.4 | 81.4 | 66.2 | 49.9 | 66.8 | 81.0 | 65.0 | 47.2 | 65.5 |
| LEMMA [19] | 73.8 | 65.8 | 48.6 | 63.5 | 83.3 | 69.5 | 52.7 | 69.4 | 81.7 | 68.3 | 49.9 | 67.6 |
| PEAN | **76.5** | **68.4** | **52.2** | **66.4** | **86.3** | **73.1** | **56.5** | **72.9** | **83.9** | **71.3** | **53.2** | **70.4** |
| HR | 85.2 | 81.8 | 76.1 | 81.3 | 94.9 | 90.6 | 82.7 | 89.8 | 94.7 | 88.8 | 80.0 | 88.3 |

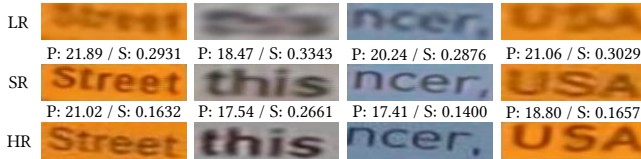

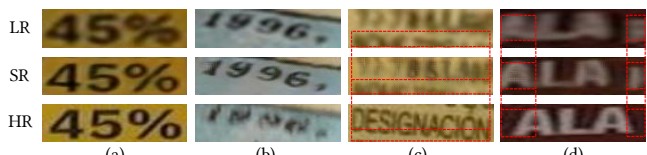

**Figure I: Visualizations about cases where SR images have lower PSNR and SSIM than LR images. "P" and "S" stand for "PSNR" and "SSIM" respectively.**

**Figure II: Visualizations of drawbacks of TextZoom.**

easier. Therefore, methods with high PSNR or SSIM values do not necessarily produce readable images, and vice versa.

Secondly, some inherent drawbacks of TextZoom [62] make it unreasonable to adopt PSNR and SSIM for evaluation if we conduct experiments on this dataset. We roughly divide them into three categories, *i.e.*, difference of the background color, low-quality HR images and cutting out of position, as presented in Figure II. Figure II(a) is a representative example of the "difference of the background color" drawback, because it is evident that the background color of LR images is quite different from that of HR images, which further results in SR images with different backgrounds than HR images. Since PSNR and SSIM are full-reference image quality assessment metrics [3], and HR images are used for reference, this difference has a huge negative impact on the values of PSNR and SSIM for SR images. Figure II(b) shows another drawback, *i.e.*, "low-quality HR images". We can find that for these image pairs, the quality of the HR image is even worse than that of the LR image. Therefore, PSNR and SSIM cannot serve as appropriate evaluation metrics because, for these images, higher PSNR and SSIM mean poorer SR results.

Another common drawback of TextZoom, namely "cutting out of position", is illustrated in Figure II(c, d). According to Wang *et al.* [62], TextZoom is constructed by cutting scene text images of different resolutions from the RealSR [4] and SRRAW [71] datasets. This manual process inevitably causes misalignment as shown in Figure II(c, d). Since HR images are used as reference images, and PSNR and SSIM are only calculated at the pixel level, their values will not be high regardless of the clarity of the SR results.

Aside from our analysis, some recent works [19, 29] on STISR also find the same issue. Guo *et al.* [19] state that their proposed LEM

concentrates more on the restoration of the character areas instead of the background areas, which occupies most of a scene text image, so there will be a reduction of PSNR and SSIM. Liu *et al.* [29] also draw a conclusion that PSNR and SSIM are only partially aligned with human perception when evaluating the quality of scene text images. In a word, we claim that the recognition accuracy of scene text recognizers is the most suitable evaluation metric for STISR. PSNR and SSIM are not reliable due to their inherent full-reference property and the intrinsic drawbacks of the TextZoom dataset. Thus, values of PSNR and SSIM are only provided for reference. Low PSNR and SSIM do not necessarily mean the model is not powerful enough in STISR.

### B.3 Additional Ablation Study

We provide more ablation studies to thoroughly analyze the effectiveness of each module we propose, thereby demonstrating the reasonableness and superiority of our proposed PEAN. Consistent with the main paper, all the experiments are conducted on TextZoom [62] and we report the recognition accuracy of ASTER [53].

#### B.3.1 *Ablation Study on the TPEM*.

**Paradigm and Network Architecture.** Our proposed TPEM is a diffusion-based module that employs an MLP-based denoising network, denoted as $f_\theta$. In contrast, many researchers in this field tend to utilize the U-Net architecture [49] as the denoising network in mainstream diffusion models [16, 47, 51, 66]. Therefore, we conduct experiments to demonstrate the suitability of the MLP architecture for processing the recognition probability sequence. Additionally, we compare the diffusion-based paradigm with the traditional regression-based paradigm. In the regression-based approach, the denoising network takes the primary text prior, denoted

**Table III: The PSNR and SSIM of some mainstream STISR methods on the three subsets of TextZoom. Best scores are bold.**

| Methods | PSNR | | | | SSIM | | | |
|---|---|---|---|---|---|---|---|---|
| | Easy | Medium | Hard | Average | Easy | Medium | Hard | Average |
| TSRN [62] | **25.07** | 18.86 | 19.71 | 21.42 | 0.8897 | 0.6676 | 0.7302 | 0.7690 |
| TSRGAN [13] | 24.22 | 19.17 | 19.99 | 21.29 | 0.8791 | 0.6770 | 0.7420 | 0.7718 |
| TBSRN [5] | 23.82 | 19.17 | 19.68 | 21.05 | 0.8660 | 0.6533 | 0.7490 | 0.7614 |
| PCAN [74] | 24.57 | 19.14 | **20.26** | 21.49 | 0.8830 | 0.6781 | 0.7475 | 0.7752 |
| TG [6] | 23.34 | **19.66** | 19.90 | 21.10 | 0.8369 | 0.6499 | 0.6986 | 0.7341 |
| TPGSR [34] | 24.35 | 18.73 | 19.93 | 21.18 | 0.8860 | 0.6784 | 0.7507 | 0.7774 |
| TATT [35] | 24.72 | 19.02 | 20.31 | **21.52** | **0.9006** | **0.6911** | **0.7703** | **0.7930** |
| C3-STISR† [75] | 21.78 | 18.49 | 19.60 | 20.05 | 0.8529 | 0.6465 | 0.7125 | 0.7432 |
| LEMMA† [19] | 23.56 | 18.94 | 19.63 | 20.86 | 0.8748 | 0.6869 | 0.7486 | 0.7754 |
| PEAN | 23.76 | 19.53 | 20.20 | 21.30 | 0.8655 | 0.6795 | 0.7287 | 0.7635 |

as $P^l$, as input and generates the ETP, denoted as $P^e$. The architectures of networks used in both paradigms are similar, with the key difference being that the only input of the network is $P^l$ under the regression-based paradigm. The results presented in Table IV indicate the following: (1) For both diffusion-based and regression-based methods, the MLP is more suitable than the U-Net for processing the recognition probability sequence. (2) The diffusion-based method outperforms the regression-based method, especially when employing MLP as the denoising network. This superiority can be attributed to the powerful distribution mapping capabilities of diffusion models [67].

**Table IV: Ablation study about the paradigm and network architecture of TPEM.**

| Methods | Easy | Medium | Hard | Average |
|---|---|---|---|---|
| Regression (U-Net) | 80.0 | 64.9 | 46.5 | 64.8 |
| Regression (MLP) | 80.9 | 65.5 | 47.2 | 65.6 |
| Diffusion (U-Net) | 79.6 | 65.3 | 46.9 | 64.9 |
| Diffusion (MLP) | **84.5** | **71.4** | **52.9** | **70.6** |

*B.3.2* **Optimization Objective for Training the TPEM**. Regarding to most of the diffusion-based models [16, 47, 48, 51], researchers typically constrain the estimated noise (denoted as $\epsilon$) through the loss function, as supported by Ho *et al.* [20], for the sake of improving sample quality. However, as discussed in § 3.2.2, in the context of the TPEM, we choose to directly estimate the ETP (denoted as $P^e$) instead of focusing on the noise for enhanced performance. In this regard, we conduct experiments with both optimization objectives, and the results are presented in Table V. We argue that selecting $P^e$ as the optimization target can yield superior performance in our proposed PEAN because it encourages the network to concentrate on the enhancement of the text prior itself. This approach also opens the possibility of introducing additional constraints on the ETP, such as the CTC loss [18] denoted as $\mathcal{L}^t_{ctc}$.

---

† Considering that the paper of C3-STISR [75] and LEMMA [19] only offers the average value, we measure the PSNR and SSIM of the publicly available pre-trained models to report values on the three subsets.

**Table V: Ablation study about the objective of optimization.**

| Objectives | Easy | Medium | Hard | Average |
|---|---|---|---|---|
| $\epsilon$ | 80.0 | 65.6 | 46.3 | 65.0 |
| $P^e$ | **84.5** | **71.4** | **52.9** | **70.6** |

**Loss Functions.** As demonstrated in § 3.2.2 of the main paper, the optimization process in the TPEM is supervised through the utilization of the MAE and CTC loss [18]. In this part, we conduct experiments to validate the choice of the loss functions. In addition to the aforementioned losses, we also introduce the Kullback-Leibler (KL) divergence loss in this experiment. Its purpose is to minimize the discrepancy between the ETP (referred to as $P^e$) and TP-HR (denoted as $P^h$). The results, as presented in Table VI, reveal the following insights: (1) The MAE loss, a fundamental component introduced in the pioneering work of diffusion models [20], proves to be essential. Combining the MAE loss with either the KL divergence loss or the CTC loss leads to an improvement in performance. (2) In our work, the introduction of the CTC loss provides an improvement in performance with **+4.8** compared with employing the MAE loss only. This combined loss results in a more refined $P^e$, which in turn plays a pivotal role in guiding the SR network to generate images with enhanced semantic accuracy.

**Table VI: Ablation study about the loss function for TPEM.**

| Loss Functions | Easy | Medium | Hard | Average |
|---|---|---|---|---|
| MAE | 80.5 | 65.8 | 48.0 | 65.8 |
| KL | 79.9 | 65.2 | 47.0 | 65.1 |
| CTC [18] | 82.0 | 69.0 | 51.5 | 68.4 |
| MAE + KL | 83.9 | 70.0 | 51.2 | 69.4 |
| MAE + CTC [18] | **84.5** | **71.4** | **52.9** | **70.6** |

**Sampling Strategy and Timestep.** To strike a balance between performance and efficiency, we exploit the sampling strategy proposed in DDIM [56] in the sampling process of the TPEM, with a timestep value of $S = 1$. As demonstrated by Song *et al.* [56], the traditional sampling strategy of the DDPM [20] with a large

number of steps ($T$ steps, where $T \gg S$) can be exceedingly time-consuming. In this section, we conduct experiments to validate the effectiveness and efficiency of our proposed PEAN with the chosen sampling strategy and timestep.

**Table VII: The performance of PEAN with the sampling strategy of the DDPM [20] under different sampling timesteps. "Duration" is the time the model takes to process an image.**

| Timesteps | Easy | Medium | Hard | Average | Duration (s) |
|---|---|---|---|---|---|
| $T = 200$ | 80.2 | 66.2 | 48.8 | 66.0 | 0.29 |
| $T = 500$ | 82.4 | 66.3 | 47.9 | 66.6 | 0.66 |
| $T = 1000$ | 80.0 | 65.7 | 48.3 | 65.7 | 1.11 |
| $T = 2000$ | 80.7 | 66.1 | 49.1 | 66.3 | 2.15 |

Initially, we perform experiments using the sampling strategy of DDPM [20] while varying the timestep, selecting four different values for our experiments. The results presented in Table VII reveal that this sampling strategy is not efficient. Subsequently, we substitute such sampling strategy with the one proposed in [56]. As demonstrated by Song *et al.* [56], with this strategy, smaller timesteps will result in equal or even superior performance, prompting us to explore five different timesteps in this set of experiments. The results showcased in Table VIII verify that this modified sampling strategy speeds up the inference phase of the model. Additionally, compared with $T = 500$, which yields the best performance for DDPM in our experiments, PEAN exhibits an improvement in performance by approximately **+4.0** in average with this kind of sampling strategy.

**Table VIII: The performance of PEAN with the sampling strategy of the DDIM [56] under different sampling timesteps. "Duration" is the time the model takes to process an image.**

| Timesteps | Easy | Medium | Hard | Average | Duration (s) |
|---|---|---|---|---|---|
| $S = 1$ | **84.5** | **71.4** | **52.9** | **70.6** | **0.04** |
| $S = 5$ | 79.7 | 65.9 | 46.8 | 65.1 | 0.09 |
| $S = 10$ | 81.7 | 69.4 | 50.6 | 68.2 | 0.10 |
| $S = 100$ | 83.2 | 69.0 | 50.7 | 68.6 | 0.19 |
| $S = 500$ | 82.4 | 68.5 | 50.5 | 68.1 | 0.68 |

Furthermore, we compare the efficiency of our proposed PEAN with other mainstream text prior-based STISR methods [19, 34, 35, 70, 75]. The results displayed in Table IX demonstrate that PEAN is on par with TPGSR [34], TATT [35] and C3-STISR [75] in terms of speed, while achieving an average performance improvement of **+6.5**. It even outperforms the two recent works, *i.e.*, LEMMA [19] and RTSRN [70], in both speed and performance. In summary, our proposed PEAN stands as an effective and efficient solution when compared to previous works.

### B.3.3 *Ablation Study on the AMM.*

**Necessity of LAM and GAM.** Strip-wise attention and its variants have found application across various computer vision tasks [11, 21, 58]. However, many of these approaches focus solely

**Table IX: The performance of the mainstream text prior-based STISR methods. "Duration" is the time the model takes to process an image.**

| Methods | Easy | Medium | Hard | Average | Duration (s) |
|---|---|---|---|---|---|
| TPGSR [34] | 78.9 | 62.7 | 44.5 | 62.8 | 0.03 |
| TATT [35] | 78.9 | 63.4 | 45.4 | 63.6 | **0.02** |
| C3-STISR [75] | 79.1 | 63.3 | 46.8 | 64.1 | 0.03 |
| LEMMA [19] | 81.1 | 66.3 | 47.4 | 66.0 | 0.07 |
| RTSRN [70] | 80.4 | 66.1 | 49.1 | 66.2 | 0.10 |
| PEAN | **84.5** | **71.4** | **52.9** | **70.6** | 0.04 |

on local horizontal and vertical attention, neglecting the incorporation of global contextual information. This study aims at justifying the indispensability of simultaneously employing both LAM and GAM in the STISR task. Table X illustrates the following key observations: (1) Upon removing both LAM and GAM, the model exhibits trivial performance. Notably, the addition of LAM improves network performance by **+1.9**, while the inclusion of GAM results in a further **+3.0** improvement. This underscores the effectiveness of the self-attention mechanism as a component of the feature extractor. (2) Utilizing LAM or GAM alone yields limited performance gains. However, the combination of LAM and GAM results in a substantial performance improvement of **+10.1**. This underlines the complementary nature of signals brought by LAM and GAM to the feature extractor. The reason is that LAM can effectively extract features on a per-character basis, while GAM facilitates interaction between characters, enhancing the ability of the model to learn the semantics of text in images.

**Table X: Analysis on the necessity of LAM and GAM.**

| LAM | GAM | Easy | Medium | Hard | Average |
|---|---|---|---|---|---|
| | | 75.5 | 60.4 | 42.5 | 60.5 |
| ✓ | | 76.8 | 62.7 | 44.6 | 62.4 |
| | ✓ | 78.5 | 63.0 | 45.8 | 63.5 |
| ✓ | ✓ | **84.5** | **71.4** | **52.9** | **70.6** |

**Number of Blocks.** In this part, we evaluate how the number of blocks in the AMM affects the performance of our model. According to results illustrated in Table XI, we find that when $N = 6$, the model achieves overall the best performance. Therefore, we empirically choose $N = 6$ as the default setting for all the experiments in the main paper and this Supplementary Material.

### B.3.4 *Ablation Study on the MTL Paradigm.*

**Features Serving as the Input of the ARM.** As shown in Figure 2 of the main paper, in the training phase, the output feature of the AMM, *i.e.*, the output of the $N^{\text{th}}$ block ($N = 6$ in our paper) is sent to the ARM. Then the ARM extracts features to perform the text recognition task in the MTL paradigm. In this part, we investigate the most proper input feature for the ARM. As presented in Table XII, we can find that: (1) Even without the ARM and the MLT paradigm, our proposed PEAN can attain the performance of 67.9 on average, surpassing the current SOTA method [29] by **+1.5**. This reveals that the ARM and the MTL paradigm employed

**Table XI: Analysis on the number of blocks in the AMM.**

| $N$ | Easy | Medium | Hard | Average |
|---|---|---|---|---|
| 1 | 81.9 | 67.4 | 48.0 | 66.8 |
| 2 | 82.8 | 67.9 | 50.6 | 68.1 |
| 3 | 83.4 | 68.0 | 50.3 | 68.3 |
| 4 | 83.8 | 70.5 | 53.1 | 70.1 |
| 5 | 84.3 | 70.4 | **53.3** | 70.3 |
| 6 | **84.5** | **71.4** | 52.9 | **70.6** |
| 7 | 84.2 | 71.0 | 52.9 | 70.3 |
| 8 | 83.1 | 69.1 | 51.2 | 68.8 |

in our proposed PEAN are not the sole components contribute to the SOTA performance. (2) The adoption of the ARM can truly facilitate the training process, bringing an additional improvement in performance by **+2.7**. (3) However, if the inappropriate feature is sent into the ARM, the performance is even worse than that without the ARM. Our experiments show that sending the output of the 6$^{th}$ block into the ARM is the best choice.

**Table XII: Ablation study on features for the input of the ARM. The first line indicates the case that we do not employ the ARM for assistance. $B_i$ denotes the $i^{th}$ block of the AMM.**

| Input of ARM | Easy | Medium | Hard | Average |
|---|---|---|---|---|
| w/o ARM | 81.4 | 68.8 | 50.7 | 67.9 |
| $B_1$ output | 79.1 | 64.6 | 47.4 | 64.7 |
| $B_2$ output | 84.2 | 69.9 | 51.5 | 69.5 |
| $B_3$ output | 80.1 | 65.9 | 45.4 | 64.9 |
| $B_4$ output | 79.2 | 64.3 | 47.1 | 64.5 |
| $B_5$ output | 80.1 | 64.7 | 45.1 | 64.4 |
| $B_6$ output | **84.5** | **71.4** | **52.9** | **70.6** |
| SRM output | 79.6 | 64.4 | 46.5 | 64.5 |

**Performance Gain from the MTL Paradigm.** Here we provide a more comprehensive comparison between the AMM and the SRB [62]. While Table 4 in our main paper primarily compares the AMM and the SRB under the MTL paradigm, we conduct an additional comparison by excluding such paradigm. The results are presented in Table XIII. Our findings indicate the following: (1) Without the MTL paradigm, the AMM still outperforms the SRB by an average of **+4.2**. The introduction of it leads to an additional improvement of **+1.7**. (2) The utilization of the MTL paradigm contributes to an improvement of performance for both the AMM and the SRB. Notably, the combination of the AMM and the MTL paradigm yields superior performance.

Given that the MTL paradigm can potentially enhance other STISR methods, we introduce it into previous text prior-based STISR methods [19, 34, 35, 70, 75]. This exploration aims to validate the uniqueness of PEAN, as simply integrating the MTL paradigm with existing approaches cannot yield significant performance improvements. In line with our proposed PEAN, we input the output of the last block of the SRB (or MSRB for RTSRN [70]) from these models into the ARM and apply the MTL paradigm during training.

**Table XIII: Comparison between the AMM and the SRB.**

| Modules | MTL | Easy | Medium | Hard | Average |
|---|---|---|---|---|---|
| SRB [62] | | 79.1 | 62.7 | 46.1 | 63.7 |
| | ✓ | 80.1 | 64.4 | 46.4 | 64.7 |
| AMM | | 81.4 | 68.8 | 50.7 | 67.9 |
| | ✓ | **84.5** | **71.4** | **52.9** | **70.6** |

The results presented in Table XIV highlight a distinctive pattern: the ARM does not consistently improve the performance of other text prior-based STISR methods. In certain instances, the inclusion of the MTL paradigm even leads to a degradation in performance. This observation underscores the idea that the MTL paradigm, which is integral to the optimization phase of our proposed PEAN, are not universally beneficial components for achieving superior performance across all the STISR methods. Its effectiveness in boosting STISR performance is realized specifically when used in conjunction with our proposed PEAN.

**Table XIV: Performance of other mainstream text prior-based models when they are equipped with the MTL paradigm.**

| Methods | MTL | Easy | Medium | Hard | Average |
|---|---|---|---|---|---|
| TPGSR [34] | | **78.9** | **62.7** | **44.5** | **62.8** |
| | ✓ | 0.01 | 0.03 | 0.01 | 0.02 |
| TATT [35] | | **78.9** | **63.4** | **45.4** | **63.6** |
| | ✓ | **78.9** | 63.3 | 44.7 | 63.4 |
| C3-STISR [75] | | 79.1 | 63.3 | **46.8** | 64.1 |
| | ✓ | **79.9** | **63.4** | 46.4 | **64.3** |
| LEMMA [19] | | **81.1** | **66.3** | **47.4** | **66.0** |
| | ✓ | 76.2 | 59.0 | 43.9 | 60.7 |
| RTSRN [70] | | **80.4** | **66.1** | **49.1** | **66.2** |
| | ✓ | 73.0 | 56.3 | 36.9 | 56.5 |

*B.3.5  **Ablation Study on the Loss Functions**.*

**Performance Gain from the SFM Loss.** In the optimization phase of PEAN, we incorporate the SFM loss [6], a loss function not utilized by previous text prior-based methods. To assess the efficacy of it, we conduct experiments by introducing it into the optimization phase of established text prior-based STISR methods [19, 34, 35, 70, 75]. This analysis aims to demonstrate that the SFM loss alone is insufficient to achieve superior performance, underscoring the necessity of developing PEAN. Considering that LEMMA [19] and RTSRN [70] employ the text-focus loss [5], a loss function working similar with the SFM loss, we deliberately abandon the text-focus loss when training these two models during our experiments. The results presented in Table XV confirm our argument, showing that simply introducing the SFM loss into existing methods does not yield significant performance improvements. Additionally, our experiments demonstrate the rationality of incorporating the SFM loss into the optimization of PEAN. This integration proves beneficial, contributing to an observable performance boost for PEAN.

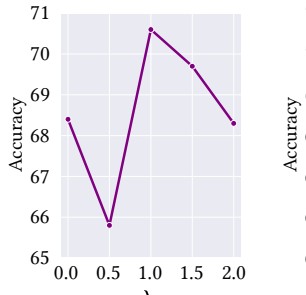 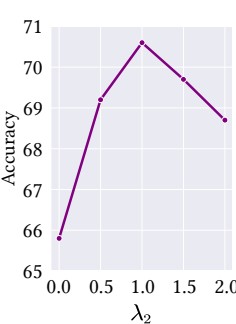 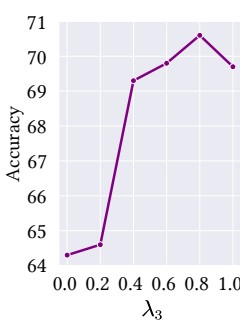 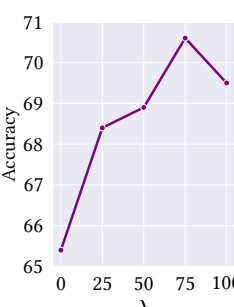 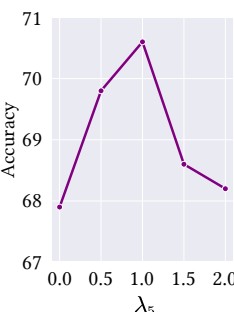

**Figure III: Ablation study on the weight of five loss functions. We report the average recognition accuracy calculated by Eq. (11).**

**Table XV: Performance of other mainstream text prior-based models when they are equipped with the SFM loss [6].**

| Methods | SFM Loss | Easy | Medium | Hard | Average |
|---|---|---|---|---|---|
| TPGSR [34] | | **78.9** | **62.7** | **44.5** | **62.8** |
| | ✓ | 74.9 | 60.1 | 41.3 | 59.8 |
| TATT [35] | | **78.9** | **63.4** | 45.4 | **63.6** |
| | ✓ | 78.3 | 62.3 | 46.3 | 63.3 |
| C3-STISR [75] | | 79.1 | 63.3 | **46.8** | 64.1 |
| | ✓ | **79.6** | **63.9** | 46.5 | **64.4** |
| LEMMA [19] | | **81.1** | **66.3** | **47.4** | **66.0** |
| | ✓ | 76.2 | 62.4 | 43.8 | 61.8 |
| RTSRN [70] | | **80.4** | **66.1** | **49.1** | **66.2** |
| | ✓ | 1.1 | 2.8 | 1.9 | 1.9 |

**Weights of the Loss Functions.** In this part, we conduct experiments to find out the best values of weights of the five loss functions, *i.e.*, $\lambda_1$ to $\lambda_5$ in the main paper. Taking weights in previous works into account [6, 62, 75], we select $\lambda_3$ in [0, 1] with an interval of 0.2. Similarly, $\lambda_4$ is selected in [0, 100] with an interval of 25. $\lambda_1$, $\lambda_2$ and $\lambda_5$ are selected in [0, 2] with an interval of 0.5. The results are presented in Figure III, from which we can find that the best combination of the weight of each loss is $\lambda_1 = 1.0$, $\lambda_2 = 1.0$, $\lambda_3 = 0.8$, $\lambda_4 = 75$ and $\lambda_5 = 1.0$. Too low or too high weights will lead to trivial performance. Therefore, we choose $\lambda_1 = 1.0$, $\lambda_2 = 1.0$, $\lambda_3 = 0.8$, $\lambda_4 = 75$ and $\lambda_5 = 1.0$ as the default setting of weights of the losses in the main paper and this Supplementary Material.

*B.3.6* **Ablation Study on Other Modules and Settings.**
**Kind of Shallow Feature Extractor.** As shown in Figure 2 of the main paper, a single convolutional layer is applied to extract the shallow feature $F^s$. Recently, CNN-Transformer-based architecture is widely adopted in SISR [28, 32, 72], so here we perform experiments to adopt CNN-Transformer-based modules as Shallow Feature Extractors. As presented in Table XVI, it is surprising to find that a single convolutional layer is enough for shallow feature extraction. Redundant ViT layers only make the model difficult to optimize and degrade the SR performance. Therefore, we use a single convolutional layer as the shallow feature extractor.
**Comparison with Other Enhancement Modules.** Since previous works such as Zhao *et al.* [75] introduce a Language Model (LM) [14] to rectify the text prior, here we compare the TPEM with

**Table XVI: Analysis on kind of the shallow feature extractor.**

| Extractors | Easy | Medium | Hard | Average |
|---|---|---|---|---|
| Conv only | **84.5** | **71.4** | **52.9** | **70.6** |
| Conv + ViT [12] | 77.9 | 62.4 | 43.7 | 62.4 |
| Conv + Swin [30] | 76.5 | 62.1 | 44.0 | 61.9 |

several LMs used in other tasks of scene text images. Among them, GSRM is an autoregressive-based LM proposed by SRN [69], while PD is the parallel Transformer decoder in PIMNet [43] based on an easy-first decoding strategy. BCN [14] is the autoencoding-based LM applied in C3-STISR [75], proposed in ABINet [14]. Experimental results are shown in Table XVII, from which we can conclude that the employment of the diffusion-based TPEM brings more performance gain compared with other variants. This can be attributed to the powerful distribution mapping capability [67] of diffusion models.

**Table XVII: Comparison with other enhancement modules.**

| Methods | Easy | Medium | Hard | Average |
|---|---|---|---|---|
| GSRM [69] | 76.3 | 58.8 | 40.7 | 59.7 |
| PD [43] | 78.1 | 61.6 | 43.4 | 62.1 |
| BCN [14] | 79.6 | 61.9 | 44.8 | 63.2 |
| PEAN | **84.5** | **71.4** | **52.9** | **70.6** |

**Performance Gain from the Pre-training Process.** As discussed in § 4.2 of the main paper, our proposed PEAN involves an initial phase where we exclude the TPEM and pre-train the model using TP-HR. Subsequently, the TPEM is introduced, and the weights of parameters obtained from the pre-training phase are initialized for the ongoing fine-tuning process. In this part, we conduct experiments aimed at investigating the impact of this configuration. We also extend this approach to established text prior-based STISR methods [19, 34, 35, 70, 75] to demonstrate that the pre-training process alone does not result in a substantial performance improvement for these methods, highlighting the necessity of proposing PEAN. The results shown in Table XIX affirm our argument. Notably, even without the pre-training process, our proposed PEAN

Table XVIII: Performance of the mainstream text prior-based models when they are equipped with different TPGs.

| Methods | CRNN [52] | ABINet [14] | PARSeq [2] | Easy | Medium | Hard | Average |
|---------|-----------|-------------|------------|------|--------|------|---------|
| | ✓ | | | 78.9 | 62.7 | 44.5 | 62.8 |
| TPGSR [34] | | ✓ | | 73.0 | 55.4 | 39.5 | 57.0 |
| | | | ✓ | 72.3 | 55.3 | 38.9 | 56.6 |
| | ✓ | | | 78.9 | 63.4 | 45.4 | 63.6 |
| TATT [35] | | ✓ | | 75.4 | 56.6 | 40.5 | 58.6 |
| | | | ✓ | 74.1 | 56.6 | 40.6 | 58.2 |
| | ✓ | | | 79.1 | 63.3 | 46.8 | 64.1 |
| C3-STISR [74] | | ✓ | | 72.5 | 54.2 | 38.8 | 56.2 |
| | | | ✓ | 75.5 | 56.7 | 38.5 | 58.1 |
| | ✓ | | | 76.1 | 58.8 | 42.7 | 60.3 |
| LEMMA [19] | | ✓ | | 81.1 | 66.3 | 47.4 | 66.0 |
| | | | ✓ | 77.6 | 60.5 | 44.7 | 62.0 |
| | ✓ | | | 80.4 | 66.1 | 49.1 | 66.2 |
| RTSRN [70] | | ✓ | | 3.3 | 2.9 | 2.2 | 2.9 |
| | | | ✓ | 80.2 | 67.5 | 46.3 | 65.7 |
| | ✓ | | | 80.8 | 66.1 | 48.6 | 66.2 |
| PEAN | | ✓ | | 82.2 | 66.0 | 47.7 | 66.4 |
| | | | ✓ | **84.5** | **71.4** | **52.9** | **70.6** |

outperforms the SOTA STISR method, *i.e.*, TextDiff [29], by an average of **+1.1**. The inclusion of the pre-training process setting leads to an additional improvement of **+3.1**.

Table XIX: Performance of the mainstream text prior-based models when equipped with the pre-training process.

| Methods | Pre-training | Easy | Medium | Hard | Average |
|---------|--------------|------|--------|------|---------|
| TPGSR [34] | | **78.9** | **62.7** | **44.5** | **62.8** |
| | ✓ | 77.6 | 61.4 | 43.6 | 61.9 |
| TATT [35] | | 78.9 | **63.4** | 45.4 | 63.6 |
| | ✓ | **79.5** | **63.4** | **45.9** | **64.0** |
| C3-STISR [75] | | **79.1** | **63.3** | **46.8** | **64.1** |
| | ✓ | 77.8 | 60.5 | 43.4 | 61.7 |
| LEMMA [19] | | 81.1 | 66.3 | 47.4 | 66.0 |
| | ✓ | **81.7** | **67.3** | **48.5** | **66.9** |
| RTSRN [70] | | **80.4** | **66.1** | **49.1** | **66.2** |
| | ✓ | 79.1 | 62.9 | 45.9 | 63.7 |
| PEAN | | 82.5 | 67.8 | 49.0 | 67.5 |
| | ✓ | **84.5** | **71.4** | **52.9** | **70.6** |

**Compatibility with different TPGs.** We adopt the pre-trained PARSeq [2] as the TPG, which is stronger than the CRNN [52] applied in [34, 35, 70, 75] and the ABINet [14] employed in [19]. For a fair comparison, we conduct experiments wherein CRNN, ABINet and PARSeq are introduced as the TPG respectively in these works. Notably, as depicted in Figure 2 of Guo *et al.* [19], LEMMA relies on the attention map sequence generated by the TPG for character location enhancement. However, CRNN is not an attention-based TPG and is unsuitable for LEMMA. To address this, we treat the output of the last convolutional layer in CRNN as the pseudo attention map and apply several linear layers to adjust its dimensions.

The results presented in Table XVIII demonstrates that our proposed PEAN exhibits compatibility with the text prior generated by CRNN, ABINet, and PARSeq. Although PARSeq [2] is more powerful than CRNN [52] and ABINet [14], previous works fail to benefit a lot from the text prior generated by it. However, with the pre-trained PARSeq as the TPG, our proposed PEAN outperforms the current SOTA text prior-based STISR method, *i.e.*, RTSRN [70] by **+4.9** on average. When ABINet is used as the TPG, RTSRN exhibits trivial performance, whereas PEAN continues to demonstrate superior performance. This indicates that our proposed PEAN has good adaptability to the text prior generated by all the three TPGs.

## C VISUALIZATIONS ON OTHER DATASETS

In this section, we provide more visualization results on the dataset we built to show the generalization of our proposed PEAN and display it ability to restore visual structure. As mentioned in the main paper, we employ IIIT5K [36], SVTP [42] and IC15 [23] for evaluation. We select 651 images whose resolution is no greater than $16 \times 64$ as LR images and input them directly into the PEAN trained on TextZoom. The results are shown in Figure IV, from which we can conclude that: (1) Previous works result in artifacts, while our proposed PEAN can address this issue. (2) Our proposed PEAN works well in terms of images with long or deformed text, while existing works tend to generate incorrect SR results.

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
