# OpenReview forum: "PEAN: A Diffusion-Based Prior-Enhanced Attention Network for Scene Text Image Super-Resolution"
_acmmm.org/ACMMM/2024/Conference — MM2024 Poster_

### Official Review · Reviewer_doDH · 2024-05-23

**Rating:** 4
**Confidence:** 3

**Summary:**

This paper proposes a novel approach for the scene text image super-resolution (STISR) task. The proposed method, PEAN, employs attention modules to accommodate various shapes and lengths of text. Additionally, the introduction of the TPEM based on a diffusion model as a prior is shown to contribute to improved accuracy.

**Strengths:**

- The manuscript is well-written and clearly structured, making it easy to understand the details of the proposed method. The authors have provided detailed explanations, especially regarding the Local Attention Module (LAM) and Global Attention Module (GAM). These sections are particularly well-explained, allowing readers to grasp the technical aspects of these components effectively.
- Personally, I found it interesting that the approach involves multi-task learning to perform both super-resolution and text recognition.
- The authors have conducted comprehensive comparisons with numerous previous studies to demonstrate the effectiveness of their approach. This extensive evaluation provides strong evidence supporting the superiority of the proposed method over existing techniques. By benchmarking against a wide range of methods, the authors convincingly show that PEAN achieves state-of-the-art performance. This thorough comparison is essential for validating the proposed method's robustness and practical applicability.
- The inclusion of an ablation study further strengthens the manuscript. The authors have meticulously dissected their method to understand the contribution of each component to the overall performance. This analysis not only reinforces the validity of the proposed approach but also provides valuable insights into the functioning of individual modules.

**Limitations:**

- How does the proposed method compare to the zero-shot performance of large-scale multimodal models? Many readers who are interested in this aspect would benefit from such an evaluation.
- While the accuracy of text recognition seems applicable in many practical scenarios, how does a task based on datasets like TextZoom compare in difficulty to real-world applications? It would be beneficial to mention the potential applications as well.

**Suitability:**

2

---

### Official Review · Reviewer_i46Z · 2024-05-24

**Rating:** 4
**Confidence:** 2

**Summary:**

This paper proposes the PEAN for scene text image super-resolution. PEAN uses an attention-based module to understand text images and a diffusion-based module to enhance text priors, improving semantic accuracy. A multi-task learning paradigm optimizes the network for generating clear SR images. PEAN achieves new state-of-the-art results on the TextZoom benchmark. Experiments show the importance of enhanced text priors in improving SR network performance.

**Strengths:**

The article is well-written and easy to follow. Compared to existing methods, it achieves state-of-the-art recognition metrics on TextZoom and PSNR and SSIM metrics on the authors' own dataset. Ablation experiments demonstrate the effectiveness of some of the designs.

**Limitations:**

The supplementary materials show that the PSNR and SSIM metrics of this method on the TextZoom dataset do not have an advantage. The authors provided a detailed analysis, explaining the shortcomings of this dataset's evaluation, which is commendable. However, given this issue, a better approach would be to correct the dataset and reevaluate the existing methods, or to use more robust image evaluation metrics such as MS-SSIM. This would make the work more beneficial to the community. Of course, this is just a friendly suggestion, not a requirement, as it goes beyond the scope of this paper.

**Suitability:**

2

---

### Official Review · Reviewer_uf7p · 2024-06-07

**Rating:** 4
**Confidence:** 3

**Summary:**

This paper focuses on the factors of visual structure and semantic information in  Scene Text Image Super-Resolution (STISR) and Scene Text Recognition (STR) problems. They proposed to process the feature map locally and globally with a strip-wise attention mechanism to recover the visual structure of images, improve semantic accuracy by utilizing a diffusion-based text enhancement module, and optimize the network with a multi-task learning paradigm.

**Strengths:**

The experiments are exhaustive and adequate, with detailed experimental explanations, and implementation details.  The ablation studies and evaluation are convincing.

**Limitations:**

The motivation part needs to be further improved and strengthened.

In line 113, the problem of "the dominance of visual structure and semantic information persists" should be elaborated. How does this problem motivate the subsequent two issues solved in this paper? Will solving these two issues solve the “dominance” problem?

For the description of the drawbacks of using Sequential Residual Blocks (SRB) and CNN-BiLSTM layers in Sec.1, the reference In line 115 includes [3, 4, 31, 56, 65] while in Sec. 2, TSRN [56] and PCAN [64] are mentioned. Is this a deliberate mismatch?

In the proposed method, CNN-BiLSTM layers are also used, doesn't they have the same "difficulty restoring the complete visual structure of images" in line 118 in Sec. 1 as well? Some of the referred works that utilize CNN-BiLSTM layers also have attention mechanisms. What makes your approach different and effective?

The motivation for the diffusion model is not clear, why can't we use TP-HR or some other ways other than TP-LR?

**Suitability:**

2

---

### Meta-Review · Area_Chair_9p4p · 2024-07-03

**Recommendation:** Accept (Poster)
**Confidence:** 4

**Metareview:**

All the reviewers provided positive rating after rebuttal. Based on the recommendation of the reviewers, the meta reviewer made the final decision.